# Unsupervised Video Summarization Based on Deep Reinforcement Learning with Interpolation

**DOI:** 10.3390/s23073384

**Published:** 2023-03-23

**Authors:** Ui Nyoung Yoon, Myung Duk Hong, Geun-Sik Jo

**Affiliations:** Artificial Intelligence Laboratory, Department of Electrical and Computer Engineering, Inha University, Incheon 22212, Republic of Korea

**Keywords:** video summarization, unsupervised learning, reinforcement learning, piecewise linear interpolation

## Abstract

Individuals spend time on online video-sharing platforms searching for videos. Video summarization helps search through many videos efficiently and quickly. In this paper, we propose an unsupervised video summarization method based on deep reinforcement learning with an interpolation method. To train the video summarization network efficiently, we used the graph-level features and designed a reinforcement learning-based video summarization framework with a temporal consistency reward function and other reward functions. Our temporal consistency reward function helped to select keyframes uniformly. We present a lightweight video summarization network with transformer and CNN networks to capture the global and local contexts to efficiently predict the keyframe-level importance score of the video in a short length. The output importance score of the network was interpolated to fit the video length. Using the predicted importance score, we calculated the reward based on the reward functions, which helped select interesting keyframes efficiently and uniformly. We evaluated the proposed method on two datasets, SumMe and TVSum. The experimental results illustrate that the proposed method showed a state-of-the-art performance compared to the latest unsupervised video summarization methods, which we demonstrate and analyze experimentally.

## 1. Introduction

Individuals spend time on online video-sharing platforms such as YouTube to search for videos. To reduce the search time, thumbnails or summary videos are used to efficiently and quickly grasp the video content [1]. Over the past few years, video summarization has become important and has been actively researched to search through video content or produce summary videos from long videos. The video summarization problem is a challenging task in predicting the frame-level or shot-level importance scores of videos [2] and is an abstract and subjective multimodal task without explicit audio-visual patterns or semantic rules. If a frame of the video is interesting or informative, the importance score of the frame should be high. These high-scored frames are selected to create the video summary. Recently, various methods that show high performance using deep learning have been proposed [3,4,5]. Deep learning-based video summarization methods are divided into supervised and unsupervised learning-based methods. For supervised learning-based methods, creating a labeled dataset is a challenge. Furthermore, it is hard to produce a dataset covering various domains or scenes. For this reason, we focused on developing an unsupervised video summarization method.

The reinforcement learning (RL) based unsupervised video summarization method proposed in [6] demonstrated an improved performance. Specifically, to train the neural network using RL, there is an efficient and explicit evaluation method to select keyframes, which is a reward function. Using the evaluation method, the deep neural network efficiently trains the various features of video such as representativeness, diversity, and uniformity. Using RL, we proposed Interp-SUM in the previous work [3], which uses the piecewise linear interpolation method. With the interpolation method, we mitigated high variance problems and improved performance with a shorter output of the network. However, since we fixed the length of output from the network, Interp-SUM had limitations in increasing the performance for long and short videos. For several videos, the keyframes were selected only in specific scenes, or the interesting keyframes were not adequately selected. Furthermore, previous RL-based video summarization methods have several weaknesses. First, it is difficult to capture the visual and temporal context with their deep neural networks. Second, many methods use the reward or loss function to train the network by calculating the visual difference among the keyframes without considering the temporal distribution of the keyframes. A summary of the video, which keeps the director’s storyline by selecting keyframes uniformly, helps people easily understand the video.

In this paper, we propose a new reinforcement learning-based video summarization framework with the interpolation method, which is composed of a new network and a new reward function such as a temporal consistency reward. To increase the performance, we used graph-level features. We also present the transformer and convolutional neural network (CNN)-based video summarization network to accurately predict the importance scores, as shown in Figure 1. The overall contributions are as follows. (i) We present a lightweight video summarization network with a transformer network and 1D convolutional neural network to capture the local and global context feature representations and for efficient interpolation. (ii) We use graph-level features as input features of the video summarization network to efficiently capture long and short context. (iii) We present the temporal consistency reward function to select interesting keyframes efficiently and uniformly.

## 2. Background and Related Work

### 2.1. Video Summarization

Video summarization methods are divided into supervised and unsupervised learning-based methods. Both methods use the video summarization dataset, which includes the frame-level or shot-level importance scores of the video annotated by several users [2,7]. The supervised learning-based method trains the model with the frame-level or shot-level features of the video as the input to predict the importance scores. With the dataset, this method calculates the cost with the difference between the predicted importance score and annotated importance score. The method minimizes the cost of finding the best model. Many supervised learning-based methods have been proposed.

In [8], the memory-augmented video summarizer was proposed. The memory network provides supporting knowledge extracted from the whole video efficiently. The global attention mechanism was used to predict the importance score of a specific shot by adjusting the score with a holistic understanding of the raw video. In [9], they presented an LSTM-based network with a determinantal point process (DPP) that encodes the probability to sample frames to learn representativeness and diversity. In [10], a dilated temporal relational (DTR) unit in the generator was presented to enhance the temporal context representation among video frames. To train the network to obtain the best summary of the video, the adversarial learning method was used with three-player loss functions. In [11], to predict the importance score to select key shots, an attention-based encoder–decoder network was proposed. This network used an encoder with a bidirectional LSTM network and a decoder with an attention mechanism to train the video representation. 

However, the issue with supervised learning-based methods is that it is very difficult to make a human-labeled video summarization dataset including videos of various categories. However, the unsupervised learning-based method does not need a human-labeled dataset. Many unsupervised learning-based methods have been proposed.

In [4], the attention autoencoder (AAE) network replaced the VAE in the SUM-GAN to improve the training efficiency and performance from the adversarial autoencoder (AAE) proposed in SUM-GAN. The interesting frames to summarize the video were weighted while training the networks. In [5], the proposed CSNet (chunk and stride network) was based on a variational autoencoder (VAE) and generative adversarial network (GAN) architecture to efficiently train the local and global contexts of the video to predict the video summary well. In [12], an adversarial autoencoder (AAE) based video summarization model was proposed. The selector LSTM selected the frames using the input frame-level features of the video. Then, the variational autoencoder (VAE) generated a reconstructed video using the selected frames. To train the entire network, the discriminator distinguishes between the original input video and the reconstructed video. In particular, four loss functions were used to train the model. In [13], the Cycle-SUM, a SUM-GAN variant, was proposed. However, the model used a cycle generative adversarial network with two VAE-based generators and two discriminators to preserve the information in the original video in the summary video. In [14], the proposed tessellation approach was a video summarization method that finds visually similar clips and selects the clips that maintain temporal coherence using the Viterbi algorithm, which is a graph-based method. Rochan et al. proposed an unsupervised learning-based SUM-FCN [15]. The method presented a new FCN architecture with the temporal convolution converted from spatial convolution to handle the video sequence. The method selects frames using the output score of the decoder and calculates the loss function with a repelling regularizer to enforce the diversity of the frames in the summary video.

### 2.2. Policy Gradient Method

Deep reinforcement learning combines a deep neural network with a reinforcement learning method [16]. The policy gradient method is one of the model-free reinforcement learning methods. The policy gradient method parametrizes the policy to the deep neural network model and optimizes the model by maximizing reward over the state distribution defined by the policy using gradient descent methods such as stochastic gradient descent (SGD). The method calculates and minimizes the cost with the objective function to train the neural network. However, policy gradient methods have several problems such as the low sample efficiency problem [17] and the high variance problem. The reason for the low sample efficiency problem is that the agent requires more samples such as human experience to learn actions in the environment (states) compared to humans as the agent is not as intelligent as a human. The high variance problem of the estimated gradient is caused by the long-horizon problem and the high-dimensional action space [18]. The long-horizon problem arises from the hugely delayed reward for a long sequence of decisions to achieve a goal. In this paper, we used a policy gradient with a baseline to reduce the variance and increase the number of episodes to mitigate the sample efficiency problem. We also used the piecewise linear interpolation to mitigate the high variance problem.

## 3. Method

The video summarization problem is formulated as a keyframe selection problem using the importance score predicted by the video summarization network. As illustrated in Figure 2, to predict accurate importance scores, we developed a video summarization network that consists of a transformer encoder network and the Pointwise Conv 1D network. The transformer encoder network encodes the input graph-level features, and Pointwise Conv 1D network decodes the features to efficiently generate importance score candidates. The importance score candidates are interpolated to the importance scores with piecewise linear interpolation and are converted to a frame-selection action to select keyframes as a summary using the Bernoulli distribution. Furthermore, the proposed temporal consistency reward function and adopted diversity and representativeness reward functions were used to measure how good the generated summary with the importance scores was. Then, we trained the video summarization network with the policy gradient-based training method using the calculated reward.

### 3.1. Video Summarization Network

First, suppose the sequence length is N and the frame number is t, the frame-level visual features ${\left\{f_{t}\right\}}_{t=1}^{N}$ are extracted from the input video using GoogleNet [19], which is a powerful CNN for image classification trained with the ImageNet dataset. Let the embedding size of the frame-level visual features is M, and the shape of the frame-level visual features is like $\left(N,\;M\right)$. Feature extraction is important for capturing the visual characteristics of the frame image as a low-dimensional feature vector. Therefore, the extracted features efficiently calculate the visual differences among frames in the video.

In recent years, graph neural networks (GNNs) have been extensively studied and it has shown state-of-the-art performance in various deep learning applications [20]. In this paper, we considered the keyframe selection problem as a graph-based anchor node finding or graph-based pathfinding problem. Because graph-level features have a relationship and the structural information of nodes, it is very useful to capture the temporal dependency among keyframes, specifically, to capture the relationship between scenes. In GNN, the graph-level features and graph representation mean that the low-dimensional node embeddings are encoded by a neural network. However, in this paper, we simply defined the graph-level features (1), that is, the features made of multiplying the node-level features $F\;\left(N\times M\right)$ and adjacency matrix $A\;\left(N\times N\right)$ without trainable embeddings, as shown in Figure 3. We built an adjacency matrix that represents edge information with node-level features.
(1)$${\left\{x_{t}\right\}}_{t=1}^{N}=FA$$

As illustrated in Figure 4a, the proposed video summarization network had a transformer and CNN to predict the keyframe-level importance score candidates with graph-level features. Transformer networks are widely used in sequence learning such as natural language understanding and video understanding because the network learns spatio-temporal context very efficiently. The original transformer network is composed of the encoder and decoder network and we only used the encoder network. The transformer encoder network in our network was made of four layers and eight heads. After the transformer encoder network, we encoded the features from 1024 to 512, which is an embedding size M using a fully connected layer. We used the layer normalization (LayerNorm) layer, which can efficiently train and distribute importance score candidate values uniformly to prevent the importance scores of each frame that do not have the same values after the sigmoid function because the sigmoid function minimizes the difference in the importance scores among frames. As illustrated in Figure 4b, the Pointwise 1D convolutional (Conv 1D) network is a very lightweight network to train temporal dependencies. It summarizes the frame-level features into the short length I of features to make importance score candidates using the convolution filter as a sliding window. In the previous work, Interp-SUM [3], since the method fixed the length I of the importance score candidate, had limitations in terms of increasing the performance for the long and short videos. On the other hand, the Pointwise Conv 1D network provides flexibility for various lengths of videos. For example, if the sequence length of a video is 213, kernel size is 15, and stride size is 3, then the length of the importance score candidates is 65 as shown in Figure 4a. After the network, we reduced the embedding size to 1 and used the sigmoid function to obtain the importance score candidates $C={\left\{c_{t}\right\}}_{t=1}^{N}$.

### 3.2. Piecewise Linear Interpolation

Interpolation is an estimation method that finds new data points based on the range of certain known data points. Specifically, piecewise linear interpolation connects the data points with the linear line and calculates intermediate data points on the line [3]. We first aligned the importance score candidates C to fit the sequence length N of the input features with the same intervals. We also interpolated the importance score candidates to the importance score S using piecewise linear interpolation. Finally, we obtained the importance scores of each frame as frame-selection probabilities to select the keyframes. The policy gradient-based reinforcement learning method has a high variance problem. The high variance problem occurs in the high-dimensional action space, for example, the frame-selection action space for video summarization. Moreover, since we used the Bernoulli distribution to select frames based on an exploration strategy, the reward of the frame-selection action changed in every step in the case of high-dimensional action space. Next, the variance of the gradient estimate calculated with a cumulated reward increased, and a high variance problem caused lower training efficiency and performance. To mitigate the high variance problem, we proposed an interpolation method. When the frames were selected using interpolated importance scores, adjacent frames had similar importance scores and were selected together. This had the effect of reducing the action space and mitigated the high variance problem. The interpolation method facilitated the generation of a natural sequence of summary frames because the near keyframes that had high importance scores were selected by each other. Moreover, the interpolation method reduced the computational complexity of the video summarization network. This makes the video summarization network learn faster because the network needs to predict only the I length of the important candidates, and not all.

To select the keyframes as a summary, the Bernoulli distribution (2) was used, which is a discrete probability distribution to convert the importance score S to the frame-selection action $A=\left\{a_{t}|a_{t}\in \left\{0,\;1\right\},\;t=1,..,N\right\}$. If the frame-selection action of a frame is equal to 1, this keyframe is selected as a summary. The Bernoulli distribution promotes the exploration of the various summaries of the video as it randomly creates variants of the frame-selection action.
(2)$$A\;\sim \;Bernoulli\left(a_{t};s_{t}\right)=\left\{\begin{array}{c} \;\;\;s_{t},\;\;\;\;\;\;for\;a_{t}=1\\ 1-s_{t},\;\;for\;a_{t}=0 \end{array}\right.$$

### 3.3. Reward Functions

To train the video summarization network efficiently, we adopted the diversity reward function and the representativeness reward function [6], and we proposed a temporal consistency reward function. The temporal consistency reward function and representativeness reward function considered the visual similarity distance and temporal similarity distance among keyframes.

The diversity reward function $R_{div}$ (3) computes the dissimilarity among the keyframes selected by the frame-selection action with the extracted features. With this reward function, the network is trained to predict the importance score to selecting diverse frames as keyframes of the summary. The summary, consisting of these keyframes, allows individuals to easily grasp the content of the video. To maintain the storyline of the video and reduce the computational complexity, we limited the temporal distance to 20 to calculate the dissimilarity among the selected keyframes. Without this limitation, even when the flashback scenes or similar scenes were far from the selected keyframe, they were ignored when selecting diverse frames.

Let the indices of the selected keyframes be $J=\{i_{k}|a_{i_{k}}=1,\;k=1,2,\ldots ,\;\left|J\right|\}$, and the diversity reward function is
(3)$$R_{div}=\frac{1}{\;J\left(\;J-1\right)}\underset{t\in \;J}{\sum }\underset{\begin{array}{c} t^{\prime }\in J\\ t\neq t^{\prime } \end{array}}{\sum }\left(1-\frac{x_{t}^{T}x_{t^{\prime }}}{\|x_{t}\|{}_{2}\|x_{t^{\prime }}\|{}_{2}}\right)\;$$

The representativeness reward function $R_{rep}$ (4) computes the similarity between the selected keyframes and all the keyframes of the video using the extracted features. With this reward function, the network is trained to predict the importance score to select keyframes of the summary that represent the video. $D_{t}$ is the distance between the selected keyframes and $s_{t}$ is the importance score of the selected keyframe.
(4)$$R_{rep}=exp\left(-\frac{1}{N}\overset{N}{\underset{t=1}{\sum }}D_{t}\times s_{t}\right)$$
(5)$$D_{t}=\underset{t^{\prime }\in J}{min}\|x_{t}-x_{t^{\prime }}\|{}_{2}$$

To create a superior summary of the video and increase performance, we applied the importance score S to the representativeness reward function, for example, $D\times s_{t}$. In particular, if the video summarization network predicts a high importance score for the representative keyframes that are selected, the distance D is decreased and the reward function returns a high reward. If the importance score of other keyframes except the representative keyframes is low, the reward function also returns a high reward for the selected keyframes. Therefore, the video summarization network is trained to minimize the average distance of the selected keyframes and to maximize the importance scores of the representative keyframes by minimizing the importance scores of other keyframes. This means that the reward is dynamically updated by applying the importance score to efficiently select more representative keyframes.
(6)$$R_{con}=\frac{1}{\log\left(\sum_{k=1}^{J}{\left(i_{k}-j_{k}\right)}^{2}/\left|J\right|\right)}$$
(7)$$j_{k}=(\underset{k\in \left|J\right|}{\arg\min}\|x_{i_{k}}-x\;\|)$$

The temporal consistency reward function $R_{con}$ (6) is proposed to efficiently and uniformly select representative shot-level keyframes. To calculate the reward function, we repeated the process to find the nearest neighbor of the selected keyframes $j_{k}$ (7) in all keyframes ${\left\{x_{t}\right\}}_{t=1}^{N}$ until the number of all keyframes $\left|J\right|,$ and we calculated the distance between $j_{k}$ and $j_{k}$. To normalize the reward, the distance was divided by $\left|J\right|,$ and the log probability was used. In the training process, the temporal consistency reward was increased by minimizing the distance.

To explain the temporal consistency reward in detail, as described in Figure 5, we calculated the similarities among the selected keyframes such as $V_{i}^{4^{\prime }}$ of $V_{i}^{summary}$ and the other keyframes of $V_{i}^{all}$. We selected the keyframes such as $V_{i}^{3}$, which were most similar to the keyframes of $V_{i}^{4^{\prime }}$, except for itself. Then, we defined the $K_{A}$ as a group of keyframes that had similar scenes around the neighbor keyframes and $K_{B}$ as a group of keyframes that had no similar scenes near the selected keyframes or had no similar scenes in the video, as described in Figure 5. The keyframes in $K_{A}$ are the representative keyframes of the surrounding keyframes. To figure out the number of keyframes included in $K_{A}$, we calculated the distance between the selected keyframe and the nearest neighbor of the selected keyframe and counted the selected keyframes that were shorter than the specified threshold. All of the keyframes were counted, except $K_{A}$, as the $K_{B}$. Depending on the storyline intended by the director, scenes similar to the keyframes included in $K_{B}$ often appear sparsely or are only shown once. This means that the summary created with the keyframes included in $K_{B}$ limits the users’ understanding of the video content because these keyframes are typically less interesting. In other words, these keyframes are not representative. Therefore, these keyframes need to be removed from the summary to increase the users’ understanding. Another advantage is that the temporal consistency reward function mitigates the problem of the excessive selection of specific scenes in the video and helps to select keyframes uniformly because the reward function helps to find local representative keyframes among neighboring keyframes, and not global representative keyframes. However, in the case of videos with large parts of static content, the proposed reward function increases the information redundancy in summary. However, the diversity reward function mitigates the redundancy problem while training.

### 3.4. Training with Policy Gradient

The quality of the generated summary was evaluated with the sum of the rewards. With the reward, the proposed video summarization network was trained as a parameterized policy $\pi_{\theta }$ using the policy gradient method. The method is a reinforcement learning method that explores a better action strategy to obtain a better summary using the gradient descent algorithm. To explore the variety of action strategies, the objective function of the exploration strategy of exploring the under-appreciated reward (UREX) method was used [21]. If the log probability $\log\;\pi_{\theta }(a_{t}\;|\;h_{t})$ of the action $\;\pi_{\theta }(a_{t}\;|\;h_{t})$ under the policy underestimates its reward $(a_{t}\;|\;h_{t})=\left(R_{rep}+R_{div}\right)/2+R_{con}$, the action will be further explored by the exploration strategy.

To calculate the objective function of UREX $O_{UREX}$, the log probabilities of the action and the rewards in episode J were used. $O_{UREX}$ is the expectation of the reward $R(a_{t}\;|\;h_{t})$, which is the sum of the rewards and the reward-augmented maximum likelihood (RAML) objective function. The set of normalized importance weights of the rewards for each episode j is computed using the softmax function to approximate the RAML objective function.
(8)$$O_{UREX}\left(\theta ;\tau \right)=E_{h\sim p\left(h_{t}\right)}\left\{\underset{a\in A}{\sum }R(a_{t}\;|\;h_{t})\right\}$$

The baseline, an essential technique for the policy gradient to reduce the variance of the gradient estimate and improve computational efficiency, was applied. The baseline is calculated as the moving average of rewards experienced thus far. To reduce the variance efficiently, we calculated the baselines per input video. As illustrated in the equation below, baseline B was computed as the sum of baseline $b_{1}$ for each input video and the baseline $b_{2}$, which was the average of all baselines for all videos. Finally, the $L_{rwd}$ is maximized as a cost to train the network.
(9)$$B=0.7\times b_{1}+0.3\times b_{2}$$
(10)$$L_{rwd}=O_{UREX}\left(\theta ;\tau \right)-B$$

### 3.5. Regularization

The regularization term $L_{reg}$ proposed in [6] was used to control the probability of selecting the keyframe using the importance score. If most of the importance scores are close to 1 or 0, the probability of selecting the wrong keyframes as a summary can be increased. For example, if the importance scores of all keyframes are 1, the video summarization network selects all keyframes as the summary. Consequently, this term was used to make the importance score closer to 0.5 while training. The number (0.5) means that this term helps to select keyframes as summaries evenly based on the exploration strategy of reinforcement learning. To avoid the rapid convergence of the importance score to 0.5 while training, 0.01 was multiplied as below.
(11)$$L_{reg}=0.01\times {\left(\frac{1}{N}\times \overset{N}{\underset{1}{\sum }}s_{t}-0.5\right)}^{2}$$

After all of the loss functions were computed, the final loss for video summarization $L_{\mathrm{summary}}$ was calculated, and the backpropagation was conducted.
(12)$$L_{summary}=L_{reg}-L_{rwd}$$

Algorithm 1 applies to the training procedure of the proposed video summarization network with the policy gradient method.
**Algorithm 1.** Training Video Summarization Network1: Input: Graph-level features of the video (xt)2: Output: Proposed network’s parameters (θ)3: 4: for the number of iterations do6:  C  ← Network(xt) % Generate importance score candidate7:  S ← Piecewise linear interpolation of C
8:  A ← Bernoulli Distribution(S)% Action A from the score S9:  % Calculate Reward Functions and A Loss Function using A and S10:  {θ} ←+ −∇Lreg−Lrwd% Minimization11:  % Update the network using the policy gradient method:12: end for

### 3.6. Generating a Video Summary

In the SumMe and TVSum datasets, we used the shot-level importance score to compare with other methods. To detect the shots of the video, the kernel temporal segmentation (KTS) method, which detects change points such as shot boundaries, is used [22]. The shot-level importance scores are calculated by averaging the frame-level importance scores in a shot. To generate the video summary, key shots were selected over the top 15% of the video length and sorted by the score. This step applies the same concept as the 0–1 Knapsack problem to maximize the importance of the summary video as described in [6].

## 4. Experiments

### 4.1. Dataset

Our proposed video summarization method was evaluated on two datasets: SumMe [2] and TVSum [7]. The SumMe dataset consists of 25 videos covering various topics such as extreme sports or airplane landings. Each video length was about 1 to 6.5 min long, and the average number of frames was 4692.8. The average number of shot changes was 29.76. The frame-level importance scores for each video were annotated. The TVSum dataset consists of 50 videos of various topics such as vlogs, news, and documentary. The shot-level importance scores for each video were annotated and the videos varied from 2 to 10 min, and the average number of frames was 7047.06. The average number of shot changes was 47.46.

### 4.2. Evaluation Setup

For a fair comparison with other methods on two datasets, the evaluation method used in [6] to compute the F-measure as a performance metric was applied. First, the precision and recall were calculated based on the result. Then, the F-measure was computed. Let G be the generated shot-level summary by our proposed method and A be the user-annotated summary in the dataset. Both the precision and recall were calculated based on the amount of temporal overlap between G and A, as seen below.
(13)$$\mathrm{Precision}=\frac{\mathrm{Duration}\;\mathrm{of}\;\mathrm{overlap}\;\mathrm{between}\;G\;\mathrm{and}\;A}{\mathrm{Duration}\;\mathrm{of}\;G}$$
(14)$$\mathrm{Recall}=\frac{\mathrm{Duration}\;\mathrm{of}\;\mathrm{overlap}\;\mathrm{between}\;G\;\mathrm{and}\;A}{\mathrm{Duration}\;\mathrm{of}\;A}$$
(15)$$F-\mathrm{measure}=\frac{2\times \;\mathrm{Precision}\;\times \;\mathrm{Recall}}{\mathrm{Precision}\;+\;\mathrm{Recall}}\times 100\%$$

The 5-fold cross-validation was used to find the performance of our method for a fair comparison. Our method was tested for five different random splits and the result of the average performance was taken. To create random splits, the videos were split into training and validation datasets. Moreover, the F-measure was computed using the validation dataset.

### 4.3. Implementation Details

The proposed video summarization network was developed using Pytorch 1.7.1 and consisted of a transformer encoder network and a Pointwise 1D Conv network. The transformer encoder network consisted of six transformer encoder layers with eight heads and 512 hidden units. Moreover, the Pointwise 1D Conv network was based on 15 kernel sizes and three stride sizes. The Adam optimizer was used to train the network with a learning rate of 0.00001 for 1000 epochs.

### 4.4. Performance Evaluation

#### 4.4.1. Quantitative Evaluation

As illustrated in Table 1, the proposed methods with different kernel sizes and stride sizes were compared to analyze the performance of a Pointwise 1D Conv network for interpolation. By changing the receptive field size using the kernel size, the importance score candidates were produced to capture the short-term or long-term context information in a video efficiently. By changing the stride size, we reduced the size of important score candidates to be interpolated to make a lightweight video summarization network and mitigate the high variance problems, as explained in the previous section. For the SumMe dataset, the method with 15 kernel sizes and three stride sizes showed the highest performance. For the TVSum dataset, the method with five kernel sizes and three stride sizes showed the highest performance. 

We noted that the performance was decreased because of the lack of context information caused by the shorter importance score candidates as the kernel size and stride size increased. In addition, the performance decreased rapidly as the stride size increased. We observed that the performance degradation was caused by the missed context information between scenes due to the short length of the importance score candidates because the length of the importance score candidates is mainly influenced by the stride size. The result of the TVSum dataset showed better performance than the result of the SumMe dataset at a smaller kernel size. This is because the training efficiency of the video summarization network was low since the length of the importance score candidates rapidly shortened when the kernel size was large, and the video length was long like the videos in the TVSum dataset. In other words, the training efficiency of the video summarization network was decreased by the shortened importance score candidates. We chose the proposed method as the best-performing method showing the best F-measure (51.66) on the SumMe dataset and the second best F-measure (59.86) on the TVSum dataset. We chose the best-performing method because increasing the performance of a small number of videos such as the SumMe dataset is more complicated than the TVSum dataset.

As illustrated in Table 2, variants of the proposed method were compared. Our method without the Pointwise 1D Conv network and interpolation showed lower performance on both datasets than the proposed method, which means that the proposed convolutional neural network and interpolation method are useful for improving the performance. Our method without the temporal consistency reward function showed a lower performance on the SumMe dataset, but showed a higher performance on the TVSum dataset. In the case of the SumMe dataset, the average length of the videos was short and the number of keyframes was small. The temporal consistency reward function helped to improve the performance in the dataset, which had many similar scenes among the keyframes. In the case of the TVSum dataset, the temporal consistency reward function was less effective. Because the video was long and the number of keyframes was large, similar scenes among keyframes were small. Our method without graph-level features showed a lower performance result on both datasets. This means that the graph-level features are very effective in improving performance.

Table 3 illustrates the difference between our proposed method and the existing unsupervised-based state-of-the-art methods. The results demonstrate that our proposed method showed a state-of-the-art performance on the SumMe dataset and high performance on the TVSum dataset. However, AC-SUM-GAN was 1.24% better than our proposed method on the TVSum dataset and DSR-RL-GRU was 0.57% better on the TVSum dataset. Although AC-SUM-GAN and DSR-RL-GRU performed better on the TVSum dataset, AC-SUM-GAN was more complicated than the proposed method and both methods could not make a natural sequence of summary like the proposed method without the interpolation method. However, these methods showed a lower performance than the proposed method on the SumMe dataset. Additionally, our proposed method demonstrated significantly improved results compared with the experimental results of our previous proposal (Interp-SUM). Specifically, the results showed that the videos in the SumMe dataset were close to the average keyframe length (293) such as ‘Car over camera’ (293, 73.2%), ‘Air force one’ (300, 60.0%), and ‘Kids playing in leaves’ (213, 50.2%), which showed the highest F-measure (%). In addition, the summary results of the short videos such as ‘Fire Domino’(108, 60.2%), and ‘Paluma jump’ (172, 61.9%) showed a high F-measure (%). However, the result of long videos such as ‘Cockpit landing’ (604, 28.8%), and ‘Uncut evening flight’ (645, 19.4%) showed a low F-measure (%). We noted from our analysis that it was difficult to find representative keyframes in the case of long videos, as selecting keyframes from the long video caused a high variance problem. However, there were cases where the video length was short, but the F-measure (%) was high such as ‘Fire Domino’ (108, 60.2%). We noted that the proposed method selected representative keyframes well when the visual dissimilarity among scenes was high.

#### 4.4.2. Qualitative Evaluation

Figure 6 is an example of the predicted importance scores by the proposed video summarization method and the video thumbnails of the keyframes, as presented in Figure 6a,c. The proposed video summarization network with the interpolation method generated a more natural sequence of a summary than the network without the interpolation method and the network selected the keyframes in the main content of the video well and uniformly. With the interpolation method, the network predicted the importance scores of the main content as being similar to the importance score of the highest important keyframe. As presented in Figure 6b,c, the method without linear interpolation did not properly select all keyframes of the main content as a summary because it is hard to predict the importance score accurately by the unsupervised learning-based method. 

## 5. Conclusions

In this paper, we proposed an unsupervised video summarization method based on deep reinforcement learning with an interpolation method. We designed a lightweight video summarization network to predict the accurate importance score candidates of keyframes and we interpolated the importance score candidates with a piecewise linear interpolation method to generate a natural sequence of summary and to mitigate the high variance problem. To train the video summarization network by the reinforcement learning method efficiently, we used graph-level features and proposed a temporal consistency reward function to select keyframes uniformly and adopted the representativeness and diversity reward functions. The experimental results illustrate that the proposed method showed state-of-the-art performance compared to the latest unsupervised video summarization methods, which we demonstrated and analyzed experimentally. Moreover, the proposed method robustly summarized various types of videos in the two datasets. However, based on the experimental results, the proposed method showed a lower performance on long videos due to the high variance problem. Conversely, the proposed method showed the best performance on the videos that were shorter than about 300 keyframes. Therefore, the proposed method is very useful for the video summarization of short-form videos.

## Figures and Tables

**Figure 1 sensors-23-03384-f001:**
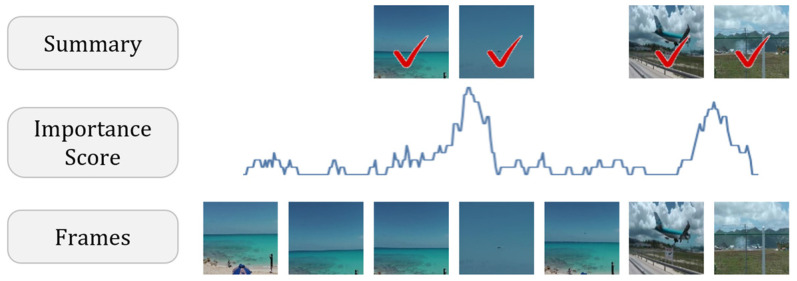
Overview: Our goal was to predict the accurate importance score of the keyframes to produce a summary.

**Figure 2 sensors-23-03384-f002:**
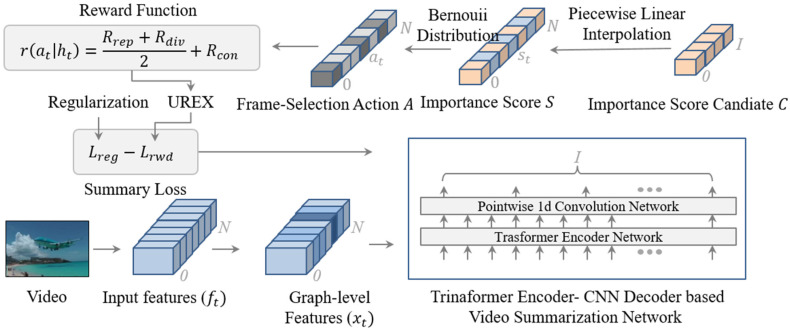
Unsupervised video summarization framework-based on deep reinforcement learning with piecewise linear interpolation.

**Figure 3 sensors-23-03384-f003:**
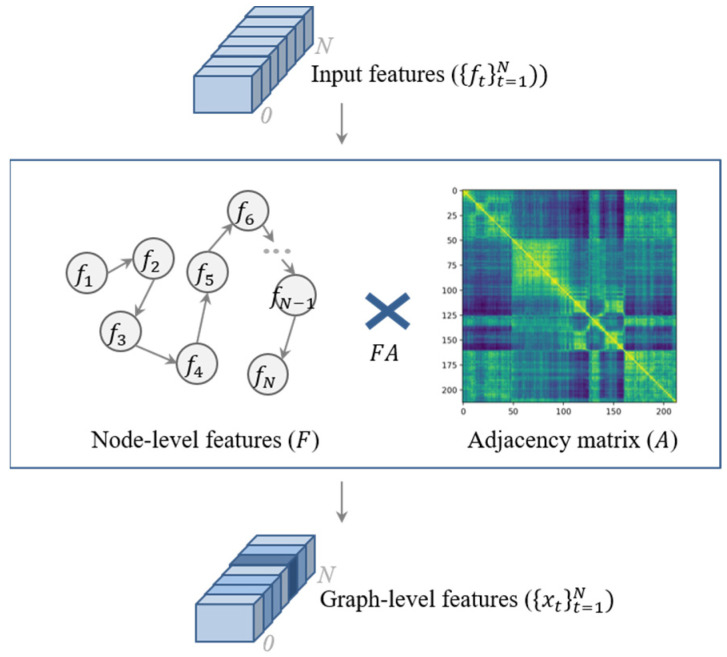
Convert the input features to the graph-level features.

**Figure 4 sensors-23-03384-f004:**
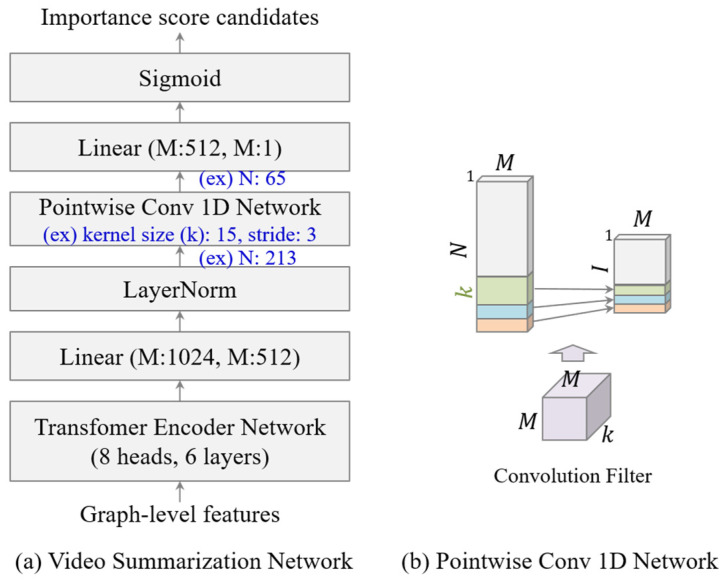
The overall architecture of our network.

**Figure 5 sensors-23-03384-f005:**
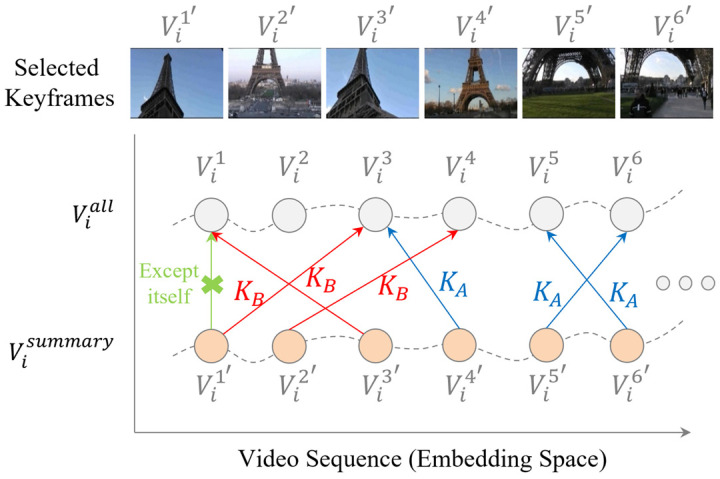
Example of the temporal consistency reward function.

**Figure 6 sensors-23-03384-f006:**
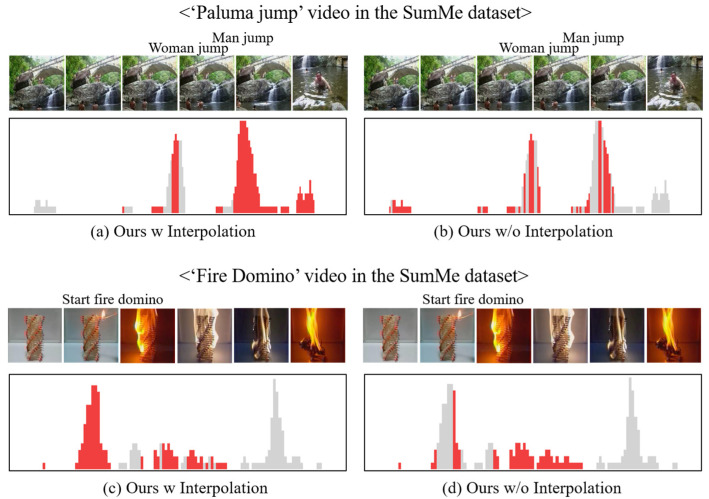
Example of the predicted importance scores by the proposed method and video thumbnails of the keyframes (The gray color means the importance score and the red color means the selected keyframes by the proposed method).

**Table 1 sensors-23-03384-t001:** Results (F-measure, %) of the comparison among our methods with different kernel and stride sizes.

Kernel Size	Stride Size	SumMe	TVSum
3	3	50.78	59.76
5	3	51.58	**60.08**
10	3	49.74	59.54
15	3	**51.66**	59.86
15	5	50.70	59.62
15	10	47.88	56.54
20	3	49.16	59.06
25	3	46.22	56.46

**Table 2 sensors-23-03384-t002:** Results (F-measure, %) of the comparison among variants with 15 kernel sizes and three stride sizes.

Method	SumMe	TVSum
Ours w/o 1D Conv Network and w/o Interpolation	50.02	58.70
Ours w/o Temporal Consistency Reward	49.88	**60.58**
Ours w/o Graph-level Features	50.46	57.26
Ours	**51.66**	59.86

**Table 3 sensors-23-03384-t003:** Results (F-measure, %) of the comparison among the unsupervised-based methods tested on SumMe and TVSum. +/− indicates better/worse performance than ours.

Method	SumMe	TVSum
SUM-GAN [12]	39.1 (−)	51.7 (−)
SUM-FCN [15]	41.5 (−)	52.7 (−)
DR-DSN [6]	41.4 (−)	57.6 (−)
Cycle-SUM [13]	41.9 (−)	57.6 (−)
CSNet [5]	51.3 (−)	58.8 (−)
UnpairedVSN [23]	47.5 (−)	55.6 (−)
SUM-GAN-AAE [4]	48.9 (−)	58.3 (−)
CSNet+GL+RPE [24]	50.2 (−)	59.1 (−)
AC-SUM-GAN [25]	50.8 (−)	**60.6 (+)**
DSR-RL-GRU [26]	50.3 (−)	**60.2 (+)**
AuDSN-SD [27]	47.7 (−)	59.8 (−)
Interp-SUM [3]	47.68 (−)	59.14 (−)
**Ours**	**51.66**	**59.86**

## Data Availability

Not applicable.

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
