# Peer review of "Unsupervised Video Summarization Based on Deep Reinforcement Learning with Interpolation"

_sensors, 2023, doi:10.3390/s23073384_

Round 1
Reviewer 1 Report
I read this paper with an interest. The organization and presented result are well described. The scale of experimentations and analysis are sufficient. Overall, I believe the manuscript can considered for publication.
Author Response
We proofread English of the manuscript according to the comment.
Reviewer 2 Report
This positive comments of this Paper “Unsupervised Video Summarization based on Deep Rein-forcement Learning with Interpolation” is as
· The proposed method shows state-of-the-art performance compared to the latest video summarization methods, as demonstrated and analyzed experimentally. This indicates the effectiveness and superiority of the proposed method in efficiently and accurately summarizing videos.
· The proposed method is based on unsupervised learning, which means it does not require manually annotated labels or ground-truth data to train the video summarization network. This makes the method more scalable, cost-effective, and applicable to various types of videos.
· Evaluation on diverse datasets: The proposed method is evaluated on two datasets, SumMe and TVSum, which have different characteristics and challenges, including varied video lengths, content, and annotations. This demonstrates the generalization ability and robustness of the proposed method and validates its effectiveness in summarizing different types of videos.
Where as
· Overall, I found your paper to be well-written and informative. However, I noticed some spelling mistakes throughout the text that could detract from the overall professionalism of the work. I recommend that you thoroughly proofread your paper and correct any spelling errors before submitting it for review. Additionally, you may want to consider using a spell-checking tool to help catch any mistakes that you may have missed.
· Complexity and computational cost: The proposed method involves multiple components, including graph-level features, deep reinforcement learning, and interpolation methods, which may make the method computationally expensive and difficult to implement in real-time applications or low-resource settings.
· The proposed method assumes a uniform and objective definition of keyframes and interestingness, which may not reflect the subjective and diverse preferences of different users or applications. The method may not capture the nuances and context-dependent aspects of video summarization, leading to suboptimal or biased summaries. Can it be applicable for surveillance videos?
· Is there any constraint in selecting the number of key frames?
Author Response
- We first thank you for the positive comments.
- We corrected spelling errors and proofread English of the manuscript according to the comments.
- To explain the complexity and computational cost, as you commented, deep reinforcement learning (RL) needs high computational cost and it is very complex to develop and optimize the whole architecture. Especially, to train the deep neural network with RL needs a long time because of the high variance problems. In contrast, graph-level features and interpolation don’t require much computational cost and they are easily implemented. Even though we couldn’t improve the RL-based video summarization method in this paper, we will develop a lightweight framework in the future.
- Since the proposed method is based on the unsupervised learning mechanism, we proposed the method which uses the objective and explicit reward functions with the basic ideas of video summarization such as representativeness, diversity, and uniformity to cover various types of videos. However, as you commented, the proposed unsupervised video summarization method generates suboptimal or biased summaries in many videos. Nevertheless, the proposed method showed state-of-the-art performance on two datasets. The method using these reward functions can be used for anomaly detection in surveillance videos. In particular, anomalies can be detected by learning to find a keyframe that is largely different from neighboring keyframes using the diversity reward function.
- Selecting the number of keyframes is different for the training step and testing step. In the training step, as described in chapter 3.2, we selected the number of keyframes using the Bernoulli distribution with the importance score. In the testing step, as described in chapter 3.6, we selected the number of keyframes over the top 15% of the video length and sorted by the score with the same concept as the 0-1 Knapsack problem to maximize the importance of the summary video.
Reviewer 3 Report
Recommendation: reduce the percentage of plagiarism, taking into account the attached report.

Author Response
- We proofread English of the manuscript and we improved the presentation of the manuscript according to the comments.
- We reduced the percentage of plagiarism refer to the attached report as much as possible.

Round 2
Reviewer 3 Report
The paper can be accepted in this form.